# Attitudes towards Monkeypox vaccination and predictors of vaccination intentions among the US general public

**Maike Winters** [1,2]* *, **Amyn A. Malik** [1,2]*, **Saad B. Omer** [2,3,4]

**1** Yale Institute for Global Health, New Haven, Connecticut, United States of America, **2** Yale School of Medicine, New Haven, Connecticut, United States of America, **3** Yale School of Public Health, New Haven, Connecticut, United States of America, **4** Yale School of Nursing, Orange, Connecticut, United States of America

* These authors contributed equally to this work.
* maike.winters@yale.edu

**Data Availability Statement:** All data files are available from the OSF database: https://osf.io/zxdjs/.

**Funding:** The authors received no specific funding for this work.

## Abstract

Amidst an unprecedented Monkeypox outbreak, we aimed to measure knowledge, attitudes, practices and Monkeypox vaccination intentions among the U.S. adult population. We conducted an online cross-sectional survey, representative of the U.S. adult general public in June 2022. We asked participants whether they would receive a Monkeypox vaccine, if they were recommended to do so. Participants also answered questions on their self-assessed level of Monkeypox knowledge, risk perception, perceived exaggeration of the threat, and self-efficacy around Monkeypox. Furthermore, we asked about their trusted sources of information, COVID-19 vaccination status and administered the 6-item Vaccine Trust Indicator (VTI). Survey weights were created based on age, gender and race. We analyzed predictors of Monkeypox vaccination intentions using logistic regression, adjusted for education, age, race and ethnicity. A total of 856 respondents completed the survey, of which 51% (n = 436) were female and 41% (n = 348) had a college degree or higher. If recommended, 46% of respondents intended to get vaccinated against Monkeypox, 29% would not get vaccinated and 25% did not know. Almost half the respondents (47%) found their own knowledge level about Monkeypox poor or very poor. The most trusted sources of information about the outbreak were healthcare professionals and officials, but also known doctors and researchers with a large online following. Only 24% indicated that the U.S. Centers for Disease Control and Prevention should be in charge of the outbreak response. Being vaccinated against COVID-19 was a strong predictor of intention to receive a Monkeypox if recommended (adjusted Odds Ratio (aOR) 29.2, 95% Confidence Interval (CI) 13.1–65.3). Increased risk perception was positively associated with vaccination intentions (aOR 2.6, 95% CI 1.8–3.6), scoring high on the VTI as well (5.4, 95% CI 3.2–9.1). The low levels of self-assessed knowledge, vaccination intentions and influence of COVID-19 vaccination status point to a lack of clear communication.

**Competing interests:** The authors have declared that no competing interests exist.

## Introduction

For the first time, there is community transmission of Monkeypox cases outside of endemic areas in West and Central Africa [1]. The 2022 surge of cases spurred the World Health Organization to declare a Public Health Emergency of International Concern on July 23rd, 2022 [2]. Shortly thereafter, on August 4th, the U.S. Department of Health and Human Services declared the monkeypox outbreak a public health emergency [3]. The first monkeypox cases in the U.S. were detected in May and since then, more than 28,500 cases have been reported throughout the country [4]. Data from the U.S. show that around 98% of the cases are among men, and 94% was among men who have sex with men (MSM) [5, 6].

The starting phase of infectious disease outbreaks comes with uncertainty, especially when it involves new pathogens, or pathogens that are new to the region [7]. Following the emergency declarations, there was a noticeable increase in Google searches on Monkeypox in the U.S., including questions around vaccine requirements, and whether one can die of Monkeypox [8]. Unclear or too little communication can create an information void, in which misinformation easily spreads [9]. The COVID-19 pandemic has also highlighted the need for early attention to behavioral aspects of a response [10, 11].

Fortunately, and contrary to the first year of the COVID-19 pandemic, there are vaccines available against Monkeypox: second and third generation smallpox vaccines [12–15]. The third generation vaccine, the Modified Vaccinia Ankara-Bavarian Nordic vaccine (JYNNEOS), is the primary vaccine being used in the current outbreak in the U.S. [16]. At the start of the 2022 global outbreak, their real-world effectiveness was not firmly established, as data were based on observational studies of first-generation smallpox vaccines in the Democratic Republic of the Congo in the 1980s [14, 17]. As the 2022 outbreak progressed, real-world vaccine performance data suggest that the JYNNEOS vaccine offers some protection against Monkeypox [17, 18]. For instance, in the U.S., the incidence was 14 times higher in unvaccinated males compared to males who had received at least one dose of the JYNNEOS vaccine [18]. In the U.S., the current vaccination strategy includes post-exposure prophylaxis for people who are known contacts to someone with Monkeypox. The vaccine is recommended as pre-exposure prophylaxis for those with higher risk of exposure to Monkeypox, for instance MSM [19].

While the outbreak currently predominantly affects MSM, the virus can spread outside of these groups and among the general public as well, as outbreaks in West and Central Africa have shown [20]. Emerging data show racial disparities in both cases and vaccinations, whereby Black people are overrepresented among cases but underrepresented in vaccination uptake [21]. In communication with the general public, special care should be taken to prevent stigmatization of high risk groups, such as MSM [22, 23]. To date, little is known about the general public's perceived knowledge, practices and perceptions of risk around Monkeypox. Furthermore, data are lacking on Monkeypox vaccination intentions, and whether the COVID-19 pandemic influences those intentions. While the historic drop in routine childhood immunization coverage in the world is likely due to a myriad of factors, attitudes towards COVID-19 vaccines as well as COVID-19 vaccination status may have played a role [24]. A recent study (not yet peer-reviewed) points to potential spillover effects from the COVID-19 pandemic, whereby COVID-19 vaccination status and attitudes negatively influence attitudes and vaccination intentions of other vaccines, such as routine childhood immunization [25].

We therefore aimed to survey the U.S. general public about their Monkeypox vaccination intentions if they would be recommended to do so, their knowledge and trusted sources of information and to test whether COVID-19 vaccination status, risk perception, self-efficacy and knowledge were associated with Monkeypox vaccination intentions. Our results can inform communication campaigns and behavioral interventions.

## Materials and methods

An online survey was administered in the United States. A representative sample of the U.S. adult population was recruited through market research company CloudResearch in early June 2022. Participants received a small monetary incentive after completing the survey. Participants answered questions on their awareness of the Monkeypox outbreak, knowledge, risk perceptions and self-efficacy around Monkeypox. We also asked about their trusted sources of information, preferred stakeholders to be in charge of the outbreak response, COVID-19 vaccination status, as well as intentions to receive a vaccine against Monkeypox if they would be recommended to do so. Assuming conservatively that 50% of the respondents would be willing to get vaccinated, with a margin of error of 4%, we estimated that we would need 600 respondents. We inflated this number to 800 to account for missing data. Yale University Institutional Review Board approved this study (IRB protocol number 2000032980) and participants gave their informed written consent before data was collected.

We used the question '*How would you rate your knowledge level about Monkeypox*?' as a measure of self-assessed knowledge. Respondents could answer on a 5-point Likert scale, ranging from 'very poor' to 'very good'. We categorized this into 'very poor/poor', 'neutral' and 'good/very good'.

We created a composite variable for risk perception based on six statements, which were answered on a 5-point Likert scale: '*My health will be severely damaged if I contract Monkeypox*', '*I think novel Monkeypox is more severe than COVID-19*', '*Even if I fall ill with another disease, I will not go to the hospital because of the risk of getting Monkeypox there*', '*Monkeypox will inflict serious damage in my community*', '*I am scared of Monkeypox*' and '*I am very concerned about this outbreak*'. The composite variable was the average of these questions (scale 1–5) and was analyzed as a continuous variable. Cronbach's alpha for the composite variable was 0.81.

We also measured whether respondents believed the Monkeypox threat was exaggerated ('*I believe that the government is exaggerating the threat*'). The 5-point Likert scale was categorized into 'disagree', 'neutral/I don't know' and 'agree'. The question '*I believe I can protect myself against Monkeypox*' was used to measure self-efficacy on a 5-point Likert scale, which was categorized into 'disagree', 'neutral/I don't know' and 'agree'. The Vaccine Trust Indicator (VTI) comprised six items around various aspects of vaccine trust. For instance, about trust in vaccine manufacturers and pharmaceutical companies, trust in the Ministry of Health and understanding how vaccination helps the body fight infectious diseases. The scale has been and is described in more detail elsewhere [26]. An average score was created on a scale from 0–10, which was subsequently categorized into 'low' (less than 4), 'medium' (between 4 and 7) and 'high' (7 and higher). COVID-19 vaccination status was ascertained with the question '*Have you been vaccinated against COVID-19*?', which could be answered with 'yes', 'no' and 'I don't know'. The last two options were combined. Among those who were aged older than 45 years, we asked whether they had received a smallpox vaccine ('yes', 'no', 'I don't know').

Intention to receive a Monkeypox vaccine was phrased: '*If a vaccine against Monkeypox is recommended for you, will you take the vaccine*?', to which respondents could answer 'yes', 'no' or 'I don't know'. We dichotomized this variable into 'yes' and 'no/I don't know'.

### Statistical analysis

Survey weights were added and were calculated based on age, gender and race and sourced from the American Community Survey [27]. Table 1 contains the overview of the demographics both unweighted and weighted and a comparison to the American population. Descriptive statistics were summarized. The proportion and means with their 95% Confidence Intervals (CI) were calculated for trust in various information sources, self-assessed level of knowledge,

**Table 1. Demographics of the sample.**

| | Unweighted sample N (%) | Weighted sample % | American population % |
|---|---|---|---|
| **Gender** | | | |
| Male | 410 (48) | 49 | 49 |
| Female | 436 (51) | 51 | 51 |
| Other | 10 (1) | 0 | |
| **Age (years)** | | | |
| 18–25 | 103 (12) | 12 | 9 |
| 26–35 | 168 (20) | 18 | 13 |
| 36–45 | 157 (18) | 16 | 12 |
| 46–55 | 150 (18) | 17 | 12 |
| 55+ | 278 (32) | 38 | 33 |
| **Race** | | | |
| Black/African American | 113 (13) | 13 | 12 |
| American Indian/Alaska Native | 17 (2) | 2 | 1 |
| Asian | 39 (5) | 4 | 6 |
| Native Hawaiian/Other Pacific Islander | 2 (0) | 0 | 0 |
| White | 685 (80) | 81 | 62 |
| **Ethnicity** | | | |
| Hispanic | 82 (10 | 9 | 19 |
| Non-Hispanic | 774 (90) | 91 | 82 |
| **Education** | | | |
| No high school | 27 (3) | 3 | 12 |
| High school | 248 (29) | 28 | 27 |
| Some college | 233 (27) | 28 | 20 |
| College | 230 (27) | 28 | 29 |
| Graduate/Professional | 118 (14) | 13 | 13 |

risk perception, perceived exaggeration of the threat, self-efficacy, the VTI and COVID-19 vaccination status, adjusted for the survey weights.

Logistic regression analyses were then carried out to determine the associations between intention to receive a Monkeypox vaccine if recommended and self-assessed level of knowledge, self-efficacy, VTI, risk perception, perceived exaggeration of threat and COVID-19 vaccination status. Given that the outbreak currently overwhelmingly affects men, we also tested whether there was a difference between genders in vaccination intentions. Results were reported for the crude associations, as well as adjusted for education (no high school, high school, some college, college, graduate/professional), age (18–25, 26–35, 36–45, 55+), race (Black of African American, American Indian or Alaska Native, Asian, Native Hawaiian or Other Pacific Islander and White), ethnicity (Hispanic, Non-Hispanic) and survey weights.

To test whether previous smallpox vaccination was associated with Monkeypox vaccination intentions, we restricted the sample to those aged 45 years and older, and carried out logistic regression analyses, both crude and adjusted for the demographic variables listed above. A replication dataset can be found at: https://osf.io/zxdjs/.

## Results

The sample comprised 856 participants, of which 51% (weighted %, unweighted n = 436) were female, 41% (unweighted n = 348) had a college degree or higher and 38% (unweighted n = 278) were 55 years or older, which was similar to the U.S. population, see Table 1. The

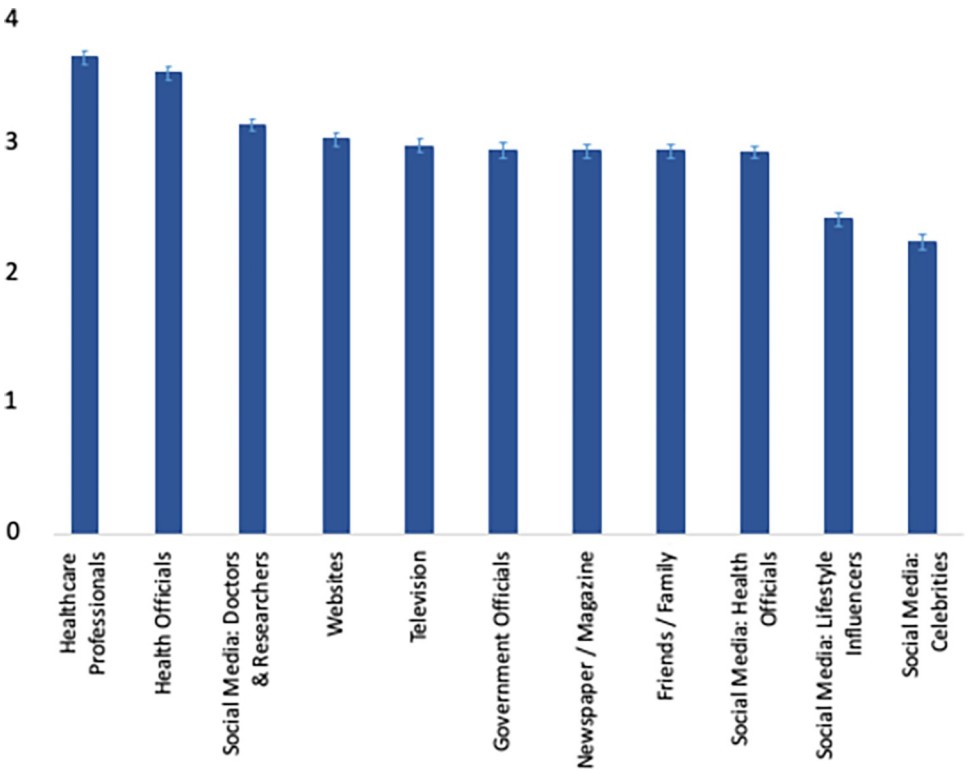

**Fig 1. Mean reliability in information sources.**

sources of information deemed most reliable to convey information about the outbreak were Healthcare Professionals (median on scale from 1–5: 3.7, standard deviation (SD) 1.4), Health Officials (like the Centers for Disease Control and Prevention (CDC)) (3.5, SD 1.3), and social media accounts of known doctors and researchers (3.1, SD 1.4), see Fig 1. Social media accounts from lifestyle influencers and celebrities scored lowest.

More than half the respondents (53%) placed most confidence in their own doctor to handle the outbreak in the US, followed by 20% who placed most confidence in the Centers for Disease Control and Prevention. Asked who should be in charge of the response, 32% ranked the President as number one, followed by the CDC (24%). While most of the respondents pointed to correct preventative measures such as avoiding close contact with sick people (83%) and washing hands with soap and water (80%), 48% said that eating a balanced diet was an effective way to prevent Monkeypox.

When asked about their COVID-19 vaccination status, 71% said they were vaccinated, which matched the share of fully vaccinated individuals in the US at the time of the survey (67%) [28]. Of the sample, 79% was aware of the Monkeypox outbreak, but almost half the respondents (47%) rated their level of knowledge about Monkeypox to be poor or very poor. 44% of the respondents were concerned about the outbreak.

More than half the respondents (54%) believed they could protect themselves against Monkeypox, only 6% believed that they could not, see Table 2. 38% of the respondents believed that the threat of Monkeypox was exaggerated by the government, 36% were neutral on this and 26% believed there was no exaggeration. On a composite variable on risk perception (see Methods), the mean score on a scale from 1–5 was 3.0 (Standard Deviation (SD) 0.8). Compared to Black and African American respondents, White respondents were less likely to

**Table 2. Monkeypox knowledge, self-efficacy, risk perception, perceived exaggeration, Vaccine Trust Indicator and COVID-19 vaccination status.**

| Indicators | % (95% Confidence Interval)* | Unweighted N |
|---|---|---|
| **Intention to get vaccinated against Monkeypox, if recommended** | | |
| No/ I don't know | 53.9 (50.1–57.6) | 482 |
| Yes | 46.1 (42.4–49.9) | 374 |
| **Self-assessed level of knowledge** | | |
| Poor / very poor | 46.7 (42.9–50.6) | 375 |
| Average | 33.8 (30.3–37.4) | 279 |
| Good / very good | 19.5 (16.5–22.8) | 144 |
| **Self-efficacy** | | |
| No self-efficacy | 6.1 (4.5–8.1) | 52 |
| Neutral | 40.2 (36.7–43.9) | 355 |
| Self-efficacy | 53.8 (50.0–57.4) | 449 |
| **Risk perception** (Mean) | 3.0 (3.0–3.1) | |
| **Perceived exaggeration of threat** | | |
| No exaggeration | 26.4 (23.2–29.8) | 225 |
| Neutral | 35.6 (32.0–39.2) | 315 |
| Exaggeration | 38.0 (34.4–41.8) | 316 |
| **Vaccine Trust Indicator** | | |
| Low | 16.2 (13.7–19.2) | 146 |
| Medium | 32.8 (29.4–36.4) | 281 |
| High | 50.9 (47.1–54.7) | 402 |
| **COVID-19 vaccination status** | | |
| Not vaccinated | 29.1 (25.7–32.6) | 255 |
| Vaccinated | 70.9 (67.4–74.3) | 581 |

*Adjusted for survey weights

perceive risk (adjusted coefficient of linear regression -0.30, 95% Confidence Interval (CI) -0.50- -0.09). There was no difference in risk perception between men and women or between the different age groups.

Asked whether they would receive a Monkeypox vaccine if recommended, 46% (95% CI 42%-50%) of our sample said 'yes', 29% (95% CI 25%-32%) declined and 25% (95% CI 22%-29%) did not know, see Table 3. Men were more likely to say yes to a Monkeypox vaccine than women (men: 57%, women 36%), Table 3. Those aged between 26–35 years had the highest vaccination intentions compared to the other age groups (55%). Vaccination intentions did not differ by race.

Looking at the predictors of willingness to receive a Monkeypox vaccine, we found that current COVID-19 vaccination status was a strong predictor (adjusted Odds Ratio (aOR) 29.3, 95% CI 13.1–65.3), see Table 4. We also found that increased risk perception was associated with Monkeypox vaccine acceptance (AOR 2.6 95% CI 1.8–3.6). Similarly, self-assessed good or very good knowledge of Monkeypox was associated with increased odds of intention to get vaccinated (aOR 2.1, 95% CI 2.2–4.0). Perceived that the threat of Monkeypox is exaggerated was in its turn negatively associated with vaccination intentions (aOR 0.4, 95% CI 0.2–0.7).

Scoring high on the Vaccine Trust Indicator (VTI), was associated with increased odds of saying yes to a Monkeypox vaccine (AOR 5.4 95% CI 3.2–0.1), compared to those who scored medium on the VTI, see Table 4. Those scoring low on the other hand were significantly less

**Table 3. Monkeypox vaccination intentions by demographics (n = 856).**

| | Intention to take Monkeypox vaccine if recommended | | | |
| --- | --- | --- | --- | --- |
| | Yes | No | I don't know | p-value |
| **Overall** | 46% | 29% | 25% | |
| **Gender** | | | | <0.000 |
| Men | 57% | 22% | 21% | |
| Women | 36% | 35% | 30% | |
| **Age** | | | | 0.033 |
| 18–25 | 44% | 33% | 22% | |
| 26–35 | 55% | 29% | 16% | |
| 36–45 | 43% | 30% | 27% | |
| 46–55 | 37% | 34% | 29% | |
| 55+ | 48% | 24% | 28% | |
| **Race** | | | | 0.390 |
| Black of African American | 44% | 25% | 31% | |
| American Indian/ Alaska Native | 54% | 41% | 5% | |
| Asian | 59% | 19% | 21% | |
| Native Hawaiian/ Other Pacific Islander | 47% | 0% | 53% | |
| White | 46% | 29% | 25% | |

**Table 4. Associations with intentions to receive a Monkeypox vaccine, if recommended.**

| | Odds Ratio (95% CI) | p-value | Adjusted* Odds Ratio (95% CI) | p-value |
| --- | --- | --- | --- | --- |
| **Self-assessed knowledge** | | | | |
| Poor / very poor | Reference | | Reference | |
| Average | 1.2 (0.8–1.7) | 0.373 | 1.1 (0.7–1.8) | 0.701 |
| Good / very good | 1.9 (1.2–2.9) | 0.005 | 2.1 (1.1–4.0) | 0.023 |
| **Self-efficacy** | | | | |
| No self-efficacy | Reference | | Reference | |
| Neutral | 0.8 (0.4–1.5) | 0.482 | 0.9 (0.3–2.4) | 0.837 |
| Self-efficacy | 1.2 (0.7–2.) | 0.536 | 0.7 (0.3–1.9) | 0.487 |
| **Risk Perception** | 1.9 (1.5–2.3) | <0.001 | 2.6 (1.8–3.6) | <0.001 |
| **Perceived exaggeration** | | | | |
| No exaggeration | Reference | | Reference | |
| Neutral | 0.3 (0.2–0.4) | <0.001 | 0.4 (0.3–0.7) | 0.002 |
| Exaggeration | 0.4 (0.3–0.6) | <0.001 | 0.4 (0.2–0.7) | 0.002 |
| **Vaccine Trust Indicator** | | | | |
| Low | 0.1 (0.0–0.3) | <0.001 | 0.3 (0.1–0.8) | 0.018 |
| Medium | Reference | | Reference | |
| High | 6.8 (4.6–10.1) | <0.001 | 5.4 (3.2–9.1) | <0.001 |
| **COVID-19 Vaccination Status** | | | | |
| Not vaccinated | Reference | | Reference | |
| Vaccinated | 29.6 (15.7–55.9) | <0.001 | 29.2 (13.1–65.3) | <0.001 |
| **Gender** | | | | |
| Male | Reference | | | |
| Female | 0.4 (0.3–0.6) | <0.001 | 0.5 (0.3–0.8) | 0.003 |

*Adjusted for: all other variables in the table, age, education, race, ethnicity, survey weights

CI = Confidence Interval

**Table 5. Variance of dependent variable explained by Vaccine Trust Indicator items.**

| Vaccine Trust Indicator item | Variance explained |
|---|---|
| Thinking about vaccination in general, would you say you are personally: 0: strongly against vaccination 10: strongly for vaccination | 4.462 |
| I generally trust vaccine manufacturers or pharmaceutical companies 0: strongly disagree 10: strongly agree | 4.805 |
| I generally trust the Department of Health & Human Services 0: strongly disagree 10: strongly agree | 4.499 |
| I understand how vaccination helps my body fight infectious disease 0: strongly disagree 10: strongly agree | 3.869 |
| I feel it is important that I get vaccinated 0: strongly disagree 10: strongly agree | 4.422 |
| Vaccination forms part of a healthy lifestyle 0: strongly disagree 10: strongly agree | 4.891 |

likely to accept a Monkeypox vaccine if recommended (AOR 0.3 95% CI 0.1–0.8). A closer look at the VTI shows that the variables 'I believe vaccination is part of a healthy lifestyle' and 'I trust vaccine manufacturers or pharmaceutical companies' explained most of the variance of the dependent variable, see Table 5.

Among those older than 45 years (55% of the sample), 67% were vaccinated against small-pox, 19% were not and 14% did not know. Those who had received a smallpox vaccine had also higher odds of saying yes to receiving a vaccine against Monkeypox compared to those who had not received a smallpox vaccine (AOR 4.7 95% CI 2.4–9.3). Similarly, those who were not aware of their smallpox vaccination status were more likely than those who had not received a smallpox vaccine to say yes to a vaccine against Monkeypox (AOR 3.6 95% CI 1.5–8.5).

## Discussion

In our representative survey of the US general population, 46% of the respondents intended to get a Monkeypox vaccine if this would be recommended to them. The low levels of self-assessed knowledge about Monkeypox are in line with general public's Monkeypox knowledge elsewhere [29] and indicate the need for clear communication about the outbreak. In recent outbreaks such as the 2014–2016 Ebola outbreak and the COVID-19 pandemic, lack of early and clear communication created space for misinformation, which in turn had a major impact on the effectiveness of outbreak control measures [30, 31].

The finding that women are less eager than men to receive a Monkeypox vaccine when rec-ommended, might be explained by the fact that the virus is currently primarily affecting MSM [32]—women might feel less need for it. However, the virus transmits through close contact and can thus transmit among women as well [32, 33]. Communication strategies could high-light the mode of transmission, as well as providing clear information on preventive methods and symptoms of Monkeypox.

Strategies should leverage trusted sources of information, such as healthcare professionals and officials, but also doctors and researchers with a large online following. In turn, these doc-tors and researchers should handle their online reach responsibly and provide accurate infor-mation to their followers [34]. The high perceived reliability in these sources of information,

as well as the large confidence our respondents had in their own doctor to handle the outbreak, has been documented during the COVID-19 pandemic as well [35]. To be able to communicate effectively about the outbreak and advise their patients, it would be important to target health workers specifically with ongoing updates around the outbreak and providing practical tools to communicate with their patients, as recent surveys (outside the US) have found that health worker knowledge about Monkeypox is suboptimal [36, 37].

The finding that COVID-19 vaccination status was strongly associated with Monkeypox vaccination intentions, can be an indicator of the influence of the pandemic experience on the general public's attitudes towards the monkeypox outbreak [38]. We also found that only 24% of the respondents felt that the CDC should be leading the outbreak response. This is in stark contrast with attitudes towards the CDC at the start of the COVID-19 pandemic, when 53% of respondents to a representative survey wanted the CDC to lead the response [35]. The pandemic has eroded trust, and efforts should be made to restore trust in the leading public health institute in the US [39].

Risk perception had a strong positive association with vaccination intentions, while believing the threat of the outbreak is exaggerated by the government was negatively associated with intentions to get vaccinated. Perceiving risk is an important determinant of behavior change [40], and can in turn also be influenced when behavior has indeed changed [41, 42]. Given the current transmission pattern of the Monkeypox outbreak, actual risks of contracting the disease vary largely, as opposed to for instance the COVID-19 pandemic [43, 44]. While our study looked at risk perception of the disease, the perceived risk of side effects and perceptions of vaccine efficacy can influence vaccination intentions as well [45].

The Vaccine Trust Indicator was strongly associated with Monkeypox vaccination intention, demonstrating the usability of this short scale. The finding that the variable around vaccination being part of a healthy lifestyle was a strong predictor, could point to a potentially powerful way of framing vaccination.

In our study, more than 67% of those aged 45 and over were vaccinated against smallpox, which, based on observational data from the Democratic Republic of the Congo in the 1980s, provides at least partial protection against Monkeypox [14]. Still, those who were vaccinated against smallpox were found to be more willing to accept a vaccine against Monkeypox. It is unclear whether those who were not vaccinated against smallpox actively refused in the past or were too young to have received it (the US stopped routine smallpox vaccination in 1972).

## Strengths & limitations

A major strength of our study was the ability to collect data from a representative sample of the US adult population, giving a rapid overview of knowledge gaps, vaccine attitudes and communication needs in a new outbreak. Limitations of the study include the fact that this is a cross-sectional survey–we cannot determine the direction of the associations we observed as reverse causality may have played a role. Furthermore, unmeasured confounders, such as occupation, may have influenced the results. While we sampled using quotas to make the sample as representative as possible to the US adult population, it may be that certain groups were over- or under-sampled. We mitigated this by adding survey weights to the analysis. Given that the Monkeypox outbreak mainly affected MSM at the time of the survey, there may have been stigma and a selection bias in our study. This may have resulted in more negative attitudes towards vaccination. Lastly, we did not ask about the sexual orientation of the participants, which could have influenced vaccination intentions.

While vaccination strategies currently target MSM [19, 46], no recommendation regarding a Monkeypox vaccine for the general public has been made. Our results highlight the need for

clear communication efforts to improve vaccination attitudes and intentions among the general public. Any future communication strategies should be responsive to the circulating rumors and misinformation.

## Author Contributions

**Conceptualization:** Amyn A. Malik, Saad B. Omer.

**Data curation:** Maike Winters, Amyn A. Malik.

**Formal analysis:** Maike Winters, Amyn A. Malik.

**Investigation:** Maike Winters.

**Methodology:** Maike Winters.

**Supervision:** Saad B. Omer.

**Writing – original draft:** Maike Winters.

**Writing – review & editing:** Maike Winters, Amyn A. Malik, Saad B. Omer.

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
