## [Decision Letter · Decision Letter 0]

3 Nov 2022

PONE-D-22-27559Attitudes towards Monkeypox vaccination and predictors of vaccination intentions among the US general publicPLOS ONE

Dear Dr. Winters,

Thank you for submitting your manuscript to PLOS ONE. After careful consideration, we feel that it has merit but does not fully meet PLOS ONE’s publication criteria as it currently stands. Therefore, we invite you to submit a revised version of the manuscript that addresses the points raised during the review process.

We look forward to receiving your revised manuscript.

Kind regards,

Harapan Harapan, MD, PhD

Academic Editor

PLOS ONE

Journal Requirements:

We will update your Data Availability statement to reflect the information you provide in your cover letter."""

Reviewers' comments:

Reviewer's Responses to Questions

**Comments to the Author**

1. Is the manuscript technically sound, and do the data support the conclusions?

Reviewer #1: Partly

Reviewer #2: Yes

2. Has the statistical analysis been performed appropriately and rigorously? 

Reviewer #1: No

Reviewer #2: Yes

3. Have the authors made all data underlying the findings in their manuscript fully available?

Reviewer #1: Yes

Reviewer #2: Yes

4. Is the manuscript presented in an intelligible fashion and written in standard English?

Reviewer #1: Yes

Reviewer #2: Yes

5. Review Comments to the Author

Reviewer #1: 1. Authors consider to assess previous “COVID-19 vaccination status” as the predictor (only those > 45 y.o. asked for previous smallpox vaccination). In the case of COVID-19, previous influenza vaccination was studied as the predictor due to the similar characteristics of these disease. Why are the reasons behind this? What about the COVID-19 vaccination coverage in the US?

2. Results from questionnaire validation should be presented per-item.

3. Why occupation is not included in the patients characteristics?

4. In Education variables; what are the differences between college and some college. And why education was not assessed as the predictor?

5. Line 252-257. Regarding the vaccination safety. This may be of interest to authors to compare the results obtained from countries in Asia, Africa, and South America. Rosiello et al 2021 Narra J 1(3):e55 doi: 10.52225/narra.v1i3.55

6. “healthy lifestyle” what is the concept that vaccination could be a part of healthy lifestyle. Kindly explain.

Reviewer #2: Thanks for the invitation to review this interesting and timely manuscript.

In the current study, Maike Winters et al. conducted a KAP study on the monkeypox vaccination using a sample of adults in the US.

The major results pointed to relatively high prevalence of vaccine hesitancy or rejection besides the poor knowledge about the emerging disease in about half of the study sample.

Overall, the study is well-written with a clear description of the methodology, and clear presentation of the results. However, the Introduction and Discussion can benefit from a thorough literature review besides the need to mention the potential study limitations.

Minor points:

1. In line 35: it is recommended not to start a sentence with a number “46% of respondents…”

2. In the Introduction, the authors can benefit from the following recent references on the topic of monkeypox prevention through vaccination:

A. https://narraj.org/main/article/view/90

B. https://www.mdpi.com/1999-4915/14/10/2155

3. In the Introduction, the authors can benefit from the following recent reference on the role of MSM as a risk group and the importance of preventing stigma and discrimination towards this group:

A. https://doi.org/10.1002/jmv.27913

B. https://doi.org/10.1002/jmv.27931

4. In the Methods, line 90: please provide more details regarding the online recruitment tool “CloudResearch”

5. In the Methods, please provide more details regarding the approach used for sample size calculation.

6. In the Discussion section, the authors can improve the discussion by checking the following recent references representing KAP studies addressing monkeypox:

A. https://www.mdpi.com/2414-6366/7/7/135

B. https://www.mdpi.com/2227-9032/10/9/1722

C. https://www.mdpi.com/2076-0817/11/8/904

7. The authors should add a limitations section to address the potential caveats of the study. E.g. possible selection bias, measurement bias, lack of data on sexual orientation of the participants.

8. The supplementary table can be moved into the main text.

Thank you!

6. PLOS authors have the option to publish the peer review history of their article (what does this mean?). If published, this will include your full peer review and any attached files.

Reviewer #1: **Yes: **Muhammad Iqhrammullah

Reviewer #2: No

---

## [Author Response · Author response to Decision Letter 0]

7 Nov 2022

We would like to thank the reviewers for their comments and suggestions on our manuscript. We have addressed each comment in the letter below, our answers are marked in blue. We have also indicated where changes in the manuscript were made. 

Dr. Maike Winters

Reviewer 1

1. Authors consider to assess previous “COVID-19 vaccination status” as the predictor (only those > 45 y.o. asked for previous smallpox vaccination). In the case of COVID-19, previous influenza vaccination was studied as the predictor due to the similar characteristics of these disease. Why are the reasons behind this? What about the COVID-19 vaccination coverage in the US?

Thank you for these questions. Given how polarizing the COVID-19 pandemic and COVID-19 vaccination has been, especially in the US where our study takes place, we hypothesized that COVID-19 vaccination status could be a predictor of vaccination attitudes and intentions of other vaccines. The recently reported substantial decrease in routine immunization coverage around the world is an example of this: even when vaccination services resumed around the world in 2021, the decrease in coverage continued (See UNICEF data: https://data.unicef.org/topic/child-health/immunization/ ). More negative attitudes towards the COVID-19 vaccines (and thus, likely not being vaccinated against COVID-19) could therefore also influence Monkeypox vaccination intentions. 

2. Results from questionnaire validation should be presented per-item.

Thank you for this comment and apologies if there is some confusion around the Vaccine Trust Indicator. This scale has been validated in previous surveys, the manuscript of which is currently under submission at a peer-reviewed journal. The aim of our study was therefore not to validate it, but merely to use it as a consolidated predictor of vaccine acceptance. 

3. Why occupation is not included in the patients characteristics?

We have not included occupation in the characteristics description, because we have not asked about this in the survey. Our socio-demographic characteristics include age, gender, education, race and ethnicity. Especially education correlates strongly with occupation - to keep the survey length manageable we opted to only ask about education. 

4. In Education variables; what are the differences between college and some college. And why education was not assessed as the predictor?

Thank you for this question. The ‘Some college’ category means that people enrolled in college, but did not graduate. The ‘College’ category means that people have graduated from college. We did not find a statistically significant association between education and Monkeypox vaccination intention, which is why we have not separately reported it in table 4. However, the adjusted associations shown in table 4 are adjusted for education, as reported in the table’s footnotes.

5. Line 252-257. Regarding the vaccination safety. This may be of interest to authors to compare the results obtained from countries in Asia, Africa, and South America. Rosiello et al 2021 Narra J 1(3):e55 doi: 10.52225/narra.v1i3.55

Thank you for sharing this interesting paper. We have added a sentence in the Discussion:

‘While our study looked at risk perception of the disease, the perceived risk of side effects and perceptions of vaccine efficacy can influence vaccination intentions as well (Rosiello 2021).’ (Lines 288-290)

6. “healthy lifestyle” what is the concept that vaccination could be a part of healthy lifestyle. Kindly explain.

This is part of the Vaccine Trust Indicator, a 6-item scale that has recently been validated (see answer to question 2). The idea of this statement around ‘healthy lifestyle’ is the notion that people have more positive attitudes towards vaccination (and are more likely to be vaccinated) when they perceive vaccination to be part of their attempts to live a healthy lifestyle.

Reviewer 2

Thanks for the invitation to review this interesting and timely manuscript.

In the current study, Maike Winters et al. conducted a KAP study on the monkeypox vaccination using a sample of adults in the US.

The major results pointed to relatively high prevalence of vaccine hesitancy or rejection besides the poor knowledge about the emerging disease in about half of the study sample.

Overall, the study is well-written with a clear description of the methodology, and clear presentation of the results. However, the Introduction and Discussion can benefit from a thorough literature review besides the need to mention the potential study limitations.

Thank you for these nice words. We have answered your questions and comments below. 

Minor points:

1. In line 35: it is recommended not to start a sentence with a number “46% of respondents…”

Thank you for noticing this, we have changed the order of the sentence:

‘If recommended, 46% of respondents intended to get vaccinated against Monkeypox,...’ (lines 35,36)

2. In the Introduction, the authors can benefit from the following recent references on the topic of monkeypox prevention through vaccination:

A. https://narraj.org/main/article/view/90

B. https://www.mdpi.com/1999-4915/14/10/2155

These are great resources, thank you for sharing. As the introduction in our manuscript has gotten slightly outdated since we submitted it, we have now updated the section on the vaccines to reflect the latest evidence on real-world effectiveness and the vaccination strategy in the US. We have added the two papers in the references. Please see lines 66-86 in the Introduction. 

3. In the Introduction, the authors can benefit from the following recent reference on the role of MSM as a risk group and the importance of preventing stigma and discrimination towards this group:

A. https://doi.org/10.1002/jmv.27913

B. https://doi.org/10.1002/jmv.27931

Thank you for these papers, we have added a sentence in the introduction on the importance of communication in preventing stigma and discrimination towards high risk groups, see lines 92-93:

‘In communication with the general public, special care should be taken to prevent stigmatization and discrimination towards high risk groups, such as MSM (Bragazzi 2022, Bragazzi 2022).’ 

4. In the Methods, line 90: please provide more details regarding the online recruitment tool “CloudResearch”

We have updated the Methods to include more information:

‘A representative sample of the U.S. adult population was recruited through market research company CloudResearch in early June 2022. Participants received a small monetary incentive after completing the survey.’ (lines 111-113) 

5. In the Methods, please provide more details regarding the approach used for sample size calculation.

To calculate our sample size, we used a conservative number of estimated vaccination intentions among the US adult population of 50%, and a margin of error of 4%. This led to a sample size of 600, which we have increased to 800 to account for missing data. This has been updated in the manuscript, see lines 118-120. 

6. In the Discussion section, the authors can improve the discussion by checking the following recent references representing KAP studies addressing monkeypox:

A. https://www.mdpi.com/2414-6366/7/7/135

B. https://www.mdpi.com/2227-9032/10/9/1722

C. https://www.mdpi.com/2076-0817/11/8/904

Thank you for sharing these papers. We have added references to the discussion. The Alsharani paper we cite in lines 248-250, as a comparison of general public’s knowledge outside of the US. 

‘The low levels of self-assessed knowledge about Monkeypox are in line with general public’s Monkeypox knowledge elsewhere (Alsharani 2022) and indicate the need for clear communication about the outbreak.’ 

The papers by Sallam et al an Ricco et all are cited in the section where we discuss the importance of keeping health workers updated (lines 268-272): 

‘To be able to communicate effectively about the outbreak and advise their patients, it would be important to target health workers specifically with ongoing updates around the outbreak and providing practical tools to communicate with their patients, as recent surveys (outside the US) have found that health worker knowledge about Monkeypox is suboptimal (Sallam 2022, Ricco 2022).’ 

7. The authors should add a limitations section to address the potential caveats of the study. E.g. possible selection bias, measurement bias, lack of data on sexual orientation of the participants.

Thank you for this comment. We have now added a section called ‘Strengths & Limitations’ to the Discussion (lines 305-316):

‘Strengths & Limitations

A major strength of our study was the ability to collect data from a representative sample of the US adult population, giving a rapid overview of knowledge gaps, vaccine attitudes and communication needs in a new outbreak. Limitations of the study include the fact that this is a cross-sectional survey – we cannot determine the direction of the associations we observed as reverse causality may have played a role. While we sampled using quotas to make the sample as representative as possible to the US adult population, it may be that certain groups were over- or under-sampled. We mitigated this by adding survey weights to the analysis. Given that the Monkeypox outbreak mainly affected MSM at the time of the survey, there may have been stigma and a selection bias in our study. This may have resulted in more negative attitudes towards vaccination. Lastly, we did not ask about the sexual orientation of the participants, which could have influenced vaccination intentions.’

8. The supplementary table can be moved into the main text.

We have moved the supplementary table to the main manuscript, where it is listed as Table 5.

Thank you!

---

## [Decision Letter · Decision Letter 1]

17 Nov 2022

PONE-D-22-27559R1Attitudes towards Monkeypox vaccination and predictors of vaccination intentions among the US general publicPLOS ONE

Dear Dr. Winters,

Thank you for submitting your manuscript to PLOS ONE. After careful consideration, we feel that it has merit but does not fully meet PLOS ONE’s publication criteria as it currently stands. Therefore, we invite you to submit a revised version of the manuscript that addresses the points raised during the review process.

We look forward to receiving your revised manuscript.

Kind regards,

Harapan Harapan, MD, PhD

Academic Editor

PLOS ONE

Journal Requirements:

Reviewers' comments:

Reviewer's Responses to Questions

**Comments to the Author**

1. If the authors have adequately addressed your comments raised in a previous round of review and you feel that this manuscript is now acceptable for publication, you may indicate that here to bypass the “Comments to the Author” section, enter your conflict of interest statement in the “Confidential to Editor” section, and submit your "Accept" recommendation.

Reviewer #1: All comments have been addressed

2. Is the manuscript technically sound, and do the data support the conclusions?

Reviewer #1: Partly

3. Has the statistical analysis been performed appropriately and rigorously? 

Reviewer #1: Yes

4. Have the authors made all data underlying the findings in their manuscript fully available?

Reviewer #1: No

5. Is the manuscript presented in an intelligible fashion and written in standard English?

Reviewer #1: Yes

6. Review Comments to the Author

Reviewer #1: Query 1. Please elaborate your response in the introduction.

Query 2. Please make a disclaimer that the validation has been carried out and will be published elsewhere. Also, please provide us the non-published supplementary file for the questionnaire validation.

Query 3. Please elaborate this (occupation as participants’ characteristic) as the research weakness since we cannot conclude whether the participants are general public or probably they are predominated by health care workers.

Query 6. Please explain why vaccination is considered a healthy lifestyle aspect in the paper.

Additional Query. Since it is lacking, please enrich the introduction with explanations about the MPXV vaccine, example of the reference: Ophinni et al. 2022. Narra J 2(3): e90 DOI: 10.52225/narra.v2i3.90

7. PLOS authors have the option to publish the peer review history of their article (what does this mean?). If published, this will include your full peer review and any attached files.

Reviewer #1: **Yes: **Muhammad Iqhrammullah

---

## [Author Response · Author response to Decision Letter 1]

18 Nov 2022

Rebuttal letter Revision 2 

We thank the reviewer for his additional comments. Our answers are in blue below. Please note that with the previous revision, we uploaded the dataset on OSF: https://osf.io/zxdjs/. 

Reviewer #1: Query 1. Please elaborate your response in the introduction.

We have added the following sentences to the introduction, lines 119-124: 

‘While the historic drop in routine childhood immunization coverage in the world is likely due to a myriad of factors, attitudes towards COVID-19 vaccines as well as COVID-19 vaccination status may have played a role (UNICEF 2021). A recent study (not yet peer-reviewed) points to the potential spillover effect from the COVID-19 pandemic, whereby COVID-19 vaccination status and attitudes negatively influence attitudes and vaccination intentions of other vaccines (Trujillo 2022).’ 

Query 2. Please make a disclaimer that the validation has been carried out and will be published elsewhere. Also, please provide us the non-published supplementary file for the questionnaire validation.

We have added to the Methods, lines 172-176:

‘The Vaccine Trust Indicator (VTI) comprised six items around various aspects of vaccine trust. For instance, about trust in vaccine manufacturers and pharmaceutical companies, trust in the Ministry of Health and understanding how vaccination helps to fight infectious diseases. The scale has been validated and is described in more detail elsewhere (Ellingson 2022).’

As the Ellingson paper is currently under submission elsewhere, we are not able to add the validation results in this paper as a supplement. 

Query 3. Please elaborate this (occupation as participants’ characteristic) as the research weakness since we cannot conclude whether the participants are general public or probably they are predominated by health care workers.

Thank you for highlighting this. We have added a sentence in the Strength & Limitations section of the Discussion on this:

‘Furthermore, unmeasured confounders, such as occupation, may have influenced the results.’ (Line 370-371). 

We do note that CloudResearch is a market research company that has provided reliable and representative data on the US general public. Their data collection methodology can be found here: https://go.cloudresearch.com/knowledge/how-does-turkprime-know-panel-demographic-data

For instance, the paper by Malik et al (see: https://www.sciencedirect.com/science/article/pii/S258953702030239X ), predicted that 67% of the US general public would get vaccinated against COVID-19, before vaccines were introduced, based on a dataset generated through CloudResearch. That number has held up very well in the real world, demonstrating the generalizability of the data. 

Query 6. Please explain why vaccination is considered a healthy lifestyle aspect in the paper.

This statement is part of the Vaccine Trust Indicator, which has been validated elsewhere, as noted under query 2. It is beyond the scope of this paper to describe the reasons why this particular statement was included in the scale. As described in our previous revision, we can assume that people have more positive attitudes towards vaccination when they perceive vaccination to be part of a healthy lifestyle.

Additional Query. Since it is lacking, please enrich the introduction with explanations about the MPXV vaccine, example of the reference: Ophinni et al. 2022. Narra J 2(3): e90 DOI: 10.52225/narra.v2i3.90

Thank you for this recommendation. This citation was already in our reference list, please see reference number 13 and relevant text as below, line 86-109:

‘Fortunately, and contrary to the first year of the COVID-19 pandemic, there are vaccines available against Monkeypox: second and third generation smallpox vaccines.(12-15) The third generation vaccine, the Modified Vaccinia Ankara-Bavarian Nordic vaccine (JYNNEOS), is the primary vaccine being used in the current outbreak in the U.S.(16) At the start of the 2022 global outbreak, their real-world effectiveness was not firmly established, as data were based on observational studies of first-generation smallpox vaccines in the Democratic Republic of the Congo in the 1980s.(14,17) As the 2022 outbreak progressed, real-world vaccine performance data suggest that the JYNNEOS vaccines offers some protection against Monkeypox.(17,18) For instance, in the U.S., the incidence was 14 times higher in unvaccinated males compared to males who had received at least one dose of the JYNNEOS vaccine.(18) In the U.S., the current vaccination strategy includes post-exposure prophylaxis for people who are known contacts to someone with Monkeypox. The vaccine is recommended as pre-exposure prophylaxis for those with higher risk of exposure to Monkeypox, for instance MSM.(19)’

---

## [Editor Report · Decision Letter 2]

21 Nov 2022

Attitudes towards Monkeypox vaccination and predictors of vaccination intentions among the US general public

PONE-D-22-27559R2

Dear Dr. Winters,

We’re pleased to inform you that your manuscript has been judged scientifically suitable for publication and will be formally accepted for publication once it meets all outstanding technical requirements.

Kind regards,

Harapan Harapan, MD, PhD

Academic Editor

PLOS ONE

Additional Editor Comments (optional):

During the final proofread please ensure:

All p-values with 0.000 should be corrected to <0.001. 

In text reference numbers should be placed before the not (.) not after the dot (.)

Title of Table 3: Please provide "(n = xxx)"

All P should be small p.
---

## [Editor Report · Acceptance letter]

23 Nov 2022

PONE-D-22-27559R2 

Attitudes towards Monkeypox vaccination and predictors of vaccination intentions among the US general public 

Dear Dr. Winters:

I'm pleased to inform you that your manuscript has been deemed suitable for publication in PLOS ONE. Congratulations! Your manuscript is now with our production department. 

Kind regards, 

on behalf of

Dr. Harapan Harapan 

Academic Editor

PLOS ONE